# Advances in Juvenile Dermatomyositis: Pathophysiology, Diagnosis, Treatment and Interstitial Lung Diseases—A Narrative Review

**DOI:** 10.3390/children11091046

**Published:** 2024-08-27

**Authors:** Ichiro Kobayashi

**Affiliations:** Center for Pediatric Allergy and Rheumatology, KKR Sapporo Medical Center, 3-40 Hiragishi 1-6, Toyohira-ku, Sapporo 060-0931, Japan; ichikoba@kkr-smc.com; Tel.: +81-11-822-1811

**Keywords:** juvenile idiopathic myopathy, juvenile dermatomyositis, myositis specific autoantibodies, type-I interferon, interstitial lung disease

## Abstract

Juvenile idiopathic inflammatory myopathy (JIIM) is a rare systemic autoimmune disease characterized by skeletal muscle weakness with or without a skin rash. Juvenile dermatomyositis (JDM) is the most common subtype of JIIM, accounting for 80% of JIIM. Recent studies identified several myositis-specific autoantibodies (MSAs) and myositis-associated autoantibodies (MAAs). Each MSA or MAA is associated with distinct clinical features and outcomes, although there are several differences in the prevalence of MSA/MAA and autoantibody–phenotype relationships between age and ethnic groups. Histopathological studies have revealed critical roles of type I interferons and vasculopathy in the development of JDM. Serological classification mostly corresponds to clinicopathological classification. Novel therapeutic agents, such as biologics and Janus kinase inhibitors (JAKi), have been developed; however, to date, there is a lack of high-level evidence. As advances in treatment have reduced the mortality rate of JIIM, recent studies have focused on medium- and long-term outcomes. However, rapidly progressive interstitial lung disease (RP-ILD) remains a major cause of death in anti-melanoma differentiation gene 5 autoantibody-positive JDM. Early diagnosis and intervention using a multi-drug regimen is critical for the treatment of RP-ILD. Rituximab and JAKi may reduce mortality in patients with JDM-associated RP-ILD refractory to conventional therapy.

## 1. Introduction

Idiopathic inflammatory myopathy (IIM) is a systemic autoimmune disease characterized by chronic muscle inflammation with or without a skin rash [1,2]. Idiopathic inflammatory myopathy comprises dermatomyositis (DM), polymyositis (PM), immune-mediated necrotizing myopathy (IMNM), inclusion body myositis, and myopathy associated with other connective tissue diseases (CTDs) (CTDM or overlap myositis) [1,2]. Idiopathic inflammatory myopathy IIMs with onset before the age of 18 years are referred to as juvenile IIM (JIIM). Analysis of a US cohort indicated that juvenile DM (JDM) was the most common type of JIIM, accounting for >80% of the cases, followed by JCTM (11.2%) and JPM (6.5%) [3]. Estimated annual incidence rates of JDM are 1.9–4.1 per million children per year in the US and UK [4,5]. 

Recent studies have identified several myositis-specific autoantibodies (MSAs) and myositis-associated autoantibodies (MAAs), each of which is associated with distinct clinical phenotypes and outcomes [1,2,6,7,8,9]. Thus, JIIM is currently classified based on their MSAs [1,2]. Among MSAs, anti-melanoma differentiation-associated gene 5 (MDA5) antibodies are associated with rapidly progressive interstitial lung disease (RP-LID), an often fatal complication in both adult and juvenile DM [10,11,12]. This article aimed to review advances in the pathophysiology, clinical phenotype related to MSAs, and treatment of the most common subtypes of JIIM, JDM, and JDM-associated ILD. 

## 2. Clinicopathological Phenotypes of JIIM

Juvenile DM is clinically characterized by symmetric proximal muscle weakness and pathognomonic skin diseases, such as Gottron’s papule/erythema and heliotrope rash [1,2,13,14]. Other skin lesions include periungual erythema, elongation of the nail epithelium, dilated nailfold capillaries, poikiloderma, and undermining ulcer [1,2,14,15]. Calcinosis and lipodystrophy are late sequelae associated with long-term morbidity [16,17]. Some patients lack apparent weakness, despite clinicopathologically confirmed skin lesions characteristic of JDM for >6 months without systemic treatment, and are currently named clinically amyopathic JDM (CAJDM). Clinically amyopathic JDM comprises amyopathic JDM, which lacks both clinical and laboratory evidence of myositis, and hypomyopathic JDM, which lacks clinical myopathy but has abnormal biochemical and/or radiological findings of myositis [18]. However, some patients lack skin rashes, despite pathologically confirmed myositis; that is, JDM *sine dermatitis* [19]. This type of JDM may be clinically indistinguishable from JPM, JIMNM, and muscular dystrophy, and requires muscle biopsy for differential diagnosis. Juvenile DM/DM is histologically characterized by perifascicular atrophy, microinfarction, and overexpression of major histocompatibility complex (MHC) class I and myxovirus resistance protein A (MxA) in the muscle fibers [20,21].

Myositis without characteristic skin diseases can be divided into the following subtypes: JPM, JIMNM, and overlapping myositis. These diseases, if negative for any MSAs, are often difficult to differentiate without muscle biopsy. Juvenile PM/PM is characterized by endomysial infiltrates of CD8^+^ T cells in affected muscles, but is much less frequent than previously estimated [22]. The pathological features of (J)IMNM include individualized necrotic myofibers of different stages scattered throughout muscle biopsy, myophagocytosis, and regeneration with minimal lymphocytic infiltration [23,24]. Molecules of MHC class I but not class II are expressed in the sarcolemma of non-necrotic muscle fibers. Deposition of the complement C5b–9 is evident on the sarcolemma but not on the vessel walls [23].

## 3. Pathophysiology

Similar to other autoimmune diseases, environmental factors are believed to trigger the onset of JDM in genetically susceptible individuals. Previously reported JDM-related environmental factors include ultraviolet B, medication, and infectious agents, such as parvovirus B19, Epstein–Barr virus, coxsackievirus, human immune deficiency virus, mycoplasma pneumoniae, and *Toxoplasma gondii* [25,26,27,28]. An association of severe acute respiratory syndrome coronavirus 2 with infection or vaccination has also been reported, but is not conclusive [28].

Genome-wide association studies have identified the human leukocyte antigen (HLA) ancestral haplotype, or the HLA A1-B8-DR3-DQ2 haplotype, as a genetic risk factor for both adult and JDM in Caucasians [29,30]. Additionally, an association with amino acid position 37 within HLA-DRB1 was observed in JDM but not in adult DM [31]. Juvenile DM-related polymorphisms are reported in the genes encoding tumor necrosis factor alpha (TNF-α) promotor, interleukin-1β (IL-1β), IL-1 receptor antagonist intron, interferon regulatory factor 5, mannose-binding lectin, chemokine (C-C motif) ligand 21, phospholipase C-like protein 1, and B lymphoid kinase [32,33,34].

Vasculopathy and endothelial damage are considered key features of JDM, as suggested by ischemic changes in histopathological studies of the muscle, such as perifascicular atrophy and microinfarction [35,36]. A recent study demonstrated increased circulating endothelial cells in active JDM, which is associated with elevated endothelial markers, such as the von Willebrand factor and P-selectin [37,38]. Routinely available markers of vasculopathy include fibrin degradation products and D-dimers [14]. Deposition of the membrane attack complex of complements and immunoglobulin suggests an immunological mechanism [35], although it is not clear whether MSAs are involved in endothelial damage.

Another feature is the involvement of plasmacytoid dendritic cells (PDCs) and their products, type I interferon [39,40]. Immunohistological studies of the affected muscle in JDM demonstrate infiltration of mainly CD4-positive cells, B cells, and macrophages; approximately half of the CD4-positive cells are CD4^+^CD123^+^CD11^+^ PDCs [41,42,43]. The expression of type I IFN is immunohistologically reflected by the expression of MxA, an IFN-inducible protein, in the affected myocytes of JDM [44,45]. Since MxA expression is absent in anti-synthetase syndrome (ASS), this syndrome may be distinct from JDM in terms of pathogenesis [46]. Type I IFN and its signature are elevated in the peripheral blood of patients with active JDM [47,48]. Type I IFN activates T and B cells, leading to the induction of autoantibodies and pro-inflammatory cytokines including interferon gamma (IFN-γ) [49]. Serum levels of soluble IL-2 receptors correlate with disease activity as a T-cell activation marker [50]. Additionally, type I IFN in combination with IFN-γ induces MHC class I expression [49]. Accumulation of MHC class I molecules results in endoplasmic reticulum stress, leading to cell death of the affected myocytes [51,52,53].

## 4. MSAs

Although MSAs are detected in 70–80% of patients with JIIM, the frequency of MSAs differs between ethnicities and age groups (Table 1) [6,7,54,55,56,57,58,59,60]. The clinical phenotype associated with each MSA is similar to that of adult IIM; however, there are several differences between age groups (Table 2) [6,7,9,54].

### 4.1. Anti-MDA5 Antibody

Anti-melanoma differentiation-associated gene 5 is a sensor of viral double-stranded RNA that induces type I IFN [10]. An in vitro study has demonstrated that anti-MDA5 autoantibody-MDA5-RNA complex induces IFN-α production in PDCs from healthy individuals via a toll-like receptor 7-dependent manner [61,62]. This autoantibody, originally known as anti-CADM-140 antibody, is associated with ILD, arthritis, and ulcerative lesions in both adults and children [6,7,9,10,11,54]. However, these autoantibodies are not necessarily associated with clinically amyopathic phenotypes in children [8,9,11,12]. A recent study in a Japanese cohort demonstrated the association of autoantibodies with fever, fatigue, and weight loss [9]. Anti-MDA5 autoantibodies are detected in 30% of Japanese patients with JIIM, but only in 7% of UK/Ireland and US cohorts [8,9,55,60]. A higher prevalence of this antibody has also been observed in adults with IIM in East Asia than that in other areas [54,63]. Additionally, ILD develops in 80% of Japanese patients with anti-MDA5 antibody-positive JDM, but only in 20% of UK/Ireland and 25.7% of US patients, indicating that the prevalence of ILD differs between ethnicities, even with the same autoantibodies [9,55,60]. Rapidly progressive ILD associated with anti-MDA5 autoantibodies is a major cause of death in adult and juvenile DM [64,65,66]. In contrast, patients without ILD or those who have overcome ILD tend to achieve drug-free remission [9,55,60].

### 4.2. Anti-Transcriptional Intermediary Factor 1 (TIF1)-γ Antibody

Anti-TIF1-γ antibodies (also known as anti-p155/140 antibodies) are detected in 16–35% of JIIM and 13–31% of adult IIM [6,7,8,9,54]. This antibody is associated with cancer-associated DM in adults, but not in children; rather, anti-TIF1-γ antibody-positive cases tend to have a chronic or recurrent course with typical skin rashes [6,7,8,9,54]. Notably, anti-TIF1-γ antibodies are often associated with a clinically amyopathic phenotype and lipodystrophy [17,67]. Recent studies have identified several autoantibodies that coexist with anti-TIF1-γ antibodies, such as autoantibodies to specificity protein 4, cell division cycle and apoptosis regulator protein 1 (CCAR1), Cip1-interacting zing finger protein, MICOS complex subunit MIC60, and F-box-like/WD repeat-containing protein [68]. The study demonstrated the association of these anti-TIF1-γ-associated antibodies with Raynaud’s phenomenon and less severe muscle disease, and anti-CCAR1 autoantibodies with panniculitis, which suggests that the heterogeneity of anti-TIF1-γ positive JDM is attributed to the presence or absence of these autoantibodies [68]. 

### 4.3. Anti-Nuclear Matrix Protein (NXP)-2 Antibody

Anti-NXP-2 antibodies (also known as anti-MJ antibodies) are detected in 15–25% of JIIM and 1–17% of adult IIM [6,7,9,54]. This autoantibody is associated with severe myopathy, high serum CK, and calcinosis [6,7,9]. A recent study demonstrated an association between antibodies and (J)DM *sine dermatitis* [19].

### 4.4. Anti-Mi-2 Antibody

Anti-Mi-2 antibodies are detected in 4–10% of JIIM and associated with characteristic skin lesions and mild to moderate muscle weakness called “classic JDM” [6,7]. The complications of ILD are rare in adult patients [69]. Anti-Mi-2 antibodies cross-react with TIF1-γ antigen [70]. When patients are positive for both anti-Mi-2 and anti-TIF1-γ antibodies, they may be considered anti-Mi-2 antibody-positive [70]. Patients with this autoantibody usually respond well to glucocorticoid (GC) therapy and have favorable outcomes [69].

### 4.5. Anti-Small Ubiquitin-like Modifier Activating Enzyme Autoantibody

Anti-SAE autoantibodies are prevalent in Caucasian patients with DM, but rare in Asians and children [54]. This autoantibody is associated with skin lesions typical of DM, which may precede myositis. 

### 4.6. Anti-Aminoacyl-tRNA Synthetase (ARS) Antibody

Anti-ARS antibodies include anti-Jo-1, anti-PL-7, anti-PL-12, anti-OJ, anti-EJ, anti-KS, anti-Zo, and anti-Ha antibodies [7,54]. Patients with these autoantibodies share common clinical features, such as Raynaud’s phenomenon, ILD, arthritis, and mechanic’s hand, known as ASS [7]. The prevalence of anti-ARS autoantibodies is 1–3% in JIIM and 9–24% in adult IIM [7,54].

### 4.7. IMNM-Related Autoantibodies

Immune-mediated necrotizing myopathy is divided into three groups: anti-3-hydroxy-3-methylglutaryl-coenzyme A reductase (HMGCR) antibody-positive, anti-signal recognition particle (SRP) antibody-positive, and seronegative [23,24]. Injection of anti-SRP and anti-HMGCR autoantibodies in mice induces complement-dependent myopathy, suggesting a pathogenic role for these autoantibodies [71]. In a study of a US cohort, the frequencies of anti-SRP and anti-HMGCR antibodies in JIIM were 1.1% and 1.8%, respectively, and there was a strong association between anti-HMGCR antibodies and HLA–DRB1*07:01 in children but not in adult IMNM [72]. In a Japanese study of 96 patients with JIIM, anti-SRP and anti-HMGCR antibodies were not detected in any patient [9]. In contrast, another report from East Asia demonstrated that anti-HMGCR and anti-SRP antibodies were detected in 15% and 6% of the patients with pathologically confirmed JIIM, respectively [73]. The difference in the prevalence of autoantibodies may be due to the different backgrounds of individuals; the former was a study of a pediatric rheumatology cohort, whereas the latter was from a neuropathology group. Fifteen to sixty-five percent of adult anti-HMGCR antibody-positive cases have a history of statin exposure; however, such drug exposure is rare in children [23,24].

### 4.8. MAAs

Myositis-associated autoantibodies are less specific to myositis and suggest overlap with other CTDs. A study of a US cohort demonstrated that almost 16% of patients had an MAA, including anti-Ro (6.3%), anti-U1-ribonucleoprotein (5.6%), and anti-PM-scleroderma (PM-Scl) (3.7%), which frequently coexisted with MSAs [7,74]. Several candidates for MAAs have recently been reported. Anti-heat shock cognate 71 kDa protein (HSC70) autoantibodies are recently identified anti-endothelial cell antibodies that correlate with the disease activity of JDM [75]. Anti-cytosolic 5′-nucleotidase 1A (NT5C1A) autoantibodies, which are prevalent in adult patients with inclusion body myositis, were detected in 27% of a JIIM cohort, although not specifically, and associated with severe disease [76].

## 5. Diagnosis and Classification Criteria

Bohan and Peter’s criteria have been widely used to diagnose both adult and juvenile DM/PM based on weakness, characteristic skin diseases, muscle-derived enzymes, electromyography (EMG), and muscle biopsy [13]. However, the criteria do not include current diagnostic methods and are not applicable to atypical cases, such as CADM and DM *sine dermatitis*. Additionally, needle EMG specifically detects inflammatory myopathy, but is painful and often intolerable to infants and young children.

Tanimoto’s criteria have been used for the diagnosis of both adult and juvenile PM/DM in Japan, but include EMG and only anti-ARS antibodies as MSAs, which are rarely detected in JDM [77]. We modified the Tanimoto’s criteria for juvenile cases to include magnetic resonance imaging (MRI) instead of EMG [14]. A high-intensity signal detected via T2-weighted MRI or short-tau inversion recovery is useful for the detection of muscle, fascial, and subcutaneous lesions, and accordingly, for diagnosis, decision of biopsy site, and follow-up of disease activity, although similar findings may be observed after hard exercise [14]. The modified criteria also included MSAs such as anti-MDA5, anti-NXP2, and anti-TIF1-γ antibodies in addition to anti-ARS autoantibodies [14] (Table 3). The modified criteria were used for financial support by “Action for Intractable Disease, and Specific Chronic Diseases in Childhood” in Japan.

In 2017, the European Alliance of Associations for Rheumatology and American College of Rheumatology published classification criteria for IIM, which is aimed for research use in both adults and children [78]. These criteria subclassify participants into definite, probable, and possible using a scoring system, and are applicable to patients either with or without muscle biopsy. A probability cut-off of 55% corresponding to “probable” has sensitivity/specificity of 87%/82% without biopsies and 93%/88% with biopsies in the original article [78]. A study of a European cohort confirmed high sensitivity and specificity and suggested that the inclusion of ILD improves sensitivity [79]. External validation demonstrated a sensitivity/specificity of 87.4%/92.4% in a Japanese adult cohort and 89.7%/100% in a Japanese pediatric cohort [80,81]. A study of a Turkish cohort demonstrated a sensitivity/specificity of 96.5%/85% in the total cohort, 95.8%/84.6% without muscle biopsy data, and 97%/85.7% with biopsy data in the JIIM, indicating higher sensitivity and lower specificity than the Tanimoto’s and Bohan–Peter’s criteria [82]. A Chinese study demonstrated sensitivity and specificity of 92.7% and 87.0%, respectively, in adult and juvenile IIM, which are better than the Bohan–Peter’s criteria in terms of both sensitivity and specificity [83]. An Australian group suggested that the addition of MRI and MSA/MAAs improved the specificity of the criteria [84]. Together, these criteria are highly sensitive and specific; however, they require further improvement. 

Upon the diagnosis of IIM, physicians need to know that serum levels of muscle-derived enzymes may be within the normal range in CAJDM and tend toward normal in patients untreated for a long period [18,85]. Muscle biopsy is not mandatory in (J)DM with typical skin disease, but should be performed for the diagnosis of JDM *sine dermatitis*, JPM, JIMNM, or JCTM/overlap myositis [86]. Skin biopsy is necessary in patients with suspected CAJDM [14,18]. 

## 6. Management

### 6.1. Principal of Treatment

Non-pharmacological management includes adequate physical therapy and avoidance of ultraviolet exposure [14,86]. The principle of pharmacological therapy is the early introduction of aggressive therapy to prevent prolonged disease activity, which is related to increased morbidity such as calcinosis and contracture. Glucocorticoids have dramatically reduced the mortality rate of JIIM and remain the mainstay of treatment [14,86,87,88,89]. However, GC should ideally be reduced as soon as possible to prevent serious adverse effects, such as osteoporosis, growth failure, metabolic diseases, and increased susceptibility to infection. Therefore, pediatricians tend to taper the dose of GC quickly in combination with non-GC medications, such as immunosuppressants, biologics, and intravenous immunoglobulin (IVIg) [88]. As previously described, the severity of JDM varies from one case to another. Additionally, some patients exhibit severe vascular involvement with ulcerative lesions or RP-ILD. Thus, the treatment strategy should be stratified according to severity and complications. Reflecting the high prevalence of ILD, the Scientific Research Group for Pediatric Rheumatic Diseases (SRGPRD) of Japan recommends chest computed tomography in all cases of JDM at diagnosis, regardless of the clinical severity and presence or absence of respiratory symptoms [14] (Figure 1). Because ILD often needs differential diagnosis of infectious diseases and may be complicated with cytomegalovirus (CMV) or *Pneumocystis jirovecii* infection, screening for these infections by CMV antigenemia and β-D-glucan is mandatory [14,90,91]. Additionally, because intensive immunosuppressive therapy is often indicated, screening for infectious diseases should include hepatitis B, hepatitis C, tuberculosis, and varicella [92]. Patients without ILD are evaluated for severity; those with dysphagia, dyspnea, skin ulcers, generalized edema, or gastrointestinal bleeding are classified as severe and should be managed aggressively [14]. Therapeutic strategies include both induction and maintenance therapies. This review article mainly focuses on induction therapy.

The Children’s Arthritis and Rheumatology Research Alliance (CARRA) proposed consensus treatment plans for JDM with skin-predominant disease: hydroxychloroquine (HCQ); HCQ and weekly methotrexate (MTX); and HCQ, weekly MTX, and GC. However, these plans are neither recommendations nor standard, and are used for data collection [93]. In clinical settings, topical GCs with or without tacrolimus (TAC) ointment may be the first-line therapy for amyopathic JDM [14,86,94]. A high-dose GC therapy with prednisolone or intravenous methylprednisolone (IVMP) is recommended for induction therapy in every guideline or recommendation [14,86,89,95]. Weekly MTX in combination with GC is recommended as the first-line treatment [14,86,89,95]. The CARRA proposed protocols for initial treatment plans for moderately severe JDM using MTX and prednisone in combination with or without IVMP or IVIg, and found no significant pairwise differences between the plans [95,96]. The CARRA also proposed consensus plans beyond the first two months, including timing and rate of steroid tapering [97]. Recently, the Pediatric Rheumatology International Trials Organization proposed GC tapering and discontinuation based on changes in the JDM core set measure of disease activity [98].

### 6.2. Conventional Immunosuppressants

Although cyclosporine A (CsA) has an efficacy comparable to that of MTX, the rate of adverse events is higher than that of MTX [99]. Thus, CsA is used as a second-line drug in MTX-resistant and MTX-intolerant cases. The efficacy of another calcineurin inhibitor (CNI), TAC, has also been reported in case series [100,101]. Cyclosporine A or TAC is used to treat ILD in both adult and juvenile DM, as described below. 

Intravenous cyclophosphamide (IVCYC) reduces skin, global, and muscle diseases, and has a steroid-sparing effect with no serious short-term side effects [102,103]. Intravenous cyclophosphamide combined with IVMP is recommended for severe or fulminant cases. Intravenous cyclophosphamide is used to treat RP-ILD associated with JDM.

Mycophenolate mofetil (MMF) has recently been preferred as a second-line immunosuppressive drug for JDM among CARRA members [104]. Mycophenolate mofetil is effective for both skin and muscle diseases, reduces disease activity, and has steroid-sparing effects, as reported in case series and several other case reports [105,106,107,108].

### 6.3. Intravenous Immunoglobulin

A randomized controlled study showed that IVIg therapy is safe and effective for GC-resistant muscle and skin lesions in adult patients with DM [109]. The efficacy of IVIg therapy was also demonstrated in a retrospective study involving 78 patients with JDM (30 IVIg and 48 control groups) using a marginal structural model [110]. A case series of 38 patients with JDM reported an association between adverse events and products with high IgA concentrations, despite a lack of IgA deficiency [111]. Binding to C3d and its consumption may be the mechanism of action of IVIg therapy [35,112].

### 6.4. Biologics

The use of biological agents has been reported for both adult and juvenile patients with DM. Currently, rituximab (RTX) is the most commonly used biologic, followed by abatacept (ABT), tocilizumab, and TNF inhibitors, among CARRA members [113]. The efficacy of B cell-depleting therapy with RTX in refractory JDM has been reported in several case series and control studies, particularly for skin and muscle lesions and JDM-associated ILD [114,115,116,117,118,119]. In a randomized controlled study, including 152 adult PM/DM and 48 JDM patients, 83% of the patients achieved the International Myositis Assessment and Clinical Studies Group (IMACS) preliminary definition of improvement, although the study did not meet its primary endpoint for JDM [120,121]. This study also indicated that the presence of anti-ARS and anti-Mi-2 autoantibodies, a juvenile DM subset, and reduced disease damage strongly predicted clinical improvement in patients with refractory myositis [122]. In adult DM/PM cases, high levels of myeloid type I IFN gene expression in skeletal muscles predicted responses to RTX [123].

The efficacy and safety of anti-TNF-α antibodies, such as infliximab and adalimumab, are reported in JDM for muscle and skin lesions, particularly for calcinosis [124,125]. In contrast, etanercept failed to demonstrate improvement and was associated with worsening of the disease in some patients with the TNF 308A allele [126]. A recent report suggested clinical effectiveness of direct injection of infliximab into calcinotic lesions [127].

Although the number of reports indicating the efficacy of ABT is limited [128], it is preferred next to RTX among CARRA members [113]. A case report showed the effect of ABT in recalcitrant JDM with skin ulcers and calcinosis [129]. An open-label study of abatacept involving 10 patients with JDM reported that nine patients achieved the IMACS “Definition of Improvement” in 24 weeks [130].

Recent studies demonstrated critical roles of type I IFN in the development of (J)DM [44,45,49]. Case reports have demonstrated a clinical response to anifrolumab in both adult and juvenile patients with DM having refractory skin diseases [131,132]. Phase Ib study of another anti-IFN-α monoclonal antibody, sifalimumab, demonstrated improvement in Manual Muscle Testing-8 score associated with declines in IFN-responsive gene products and soluble T-cell activation markers [133,134].

### 6.5. JAK Inhibitors

The Janus kinase (JAK)-signal transducer and activator of transcription (STAT) pathway plays a critical role in the signaling of various growth factors and cytokines, including type I and II IFN [44,47,135]. Type I and II IFN use the JAK1/Tyk2 and JAK1/JAK2 signaling pathways, respectively [49,136]. Among the currently available JAK inhibitors, tofacitinib inhibits JAK1, 2, and 3, whereas baricitinib and ruxolitinib inhibit JAK1 and 2 [136]. A recent review citing case reports and case series suggested the efficacy of JAK inhibitors in JDM patients with refractory skin and muscle lesions [137]. Notably, JAK inhibitors show clinical effects in cases refractory to aggressive therapy with MMF and/or biologics, skin ulcers, calcinosis, or ILD [138,139,140,141,142,143,144,145,146,147,148,149]. Both the short- and long-term safety and efficacy of JAK inhibitors, which are associated with a decline in serum levels of IL-1RA, have been demonstrated in retrospective studies [150,151].

### 6.6. Others

Plasma exchange and polymyxin B hemoperfusion have been reported in critically ill patients with JIIM, although a randomized controlled study failed to show the efficacy of plasma exchange and leukapheresis in adult patients with PM/DM [116,118,152,153].

A recent case series study demonstrated long-term remission after a single infusion of CD19 chimeric antigen receptor T cells (CAR-T) in autoimmune connective tissue diseases, including four refractory adult patients with IIM [154]. A case report indicated clinical remission of adult ASS-associated ILD with CAR-T therapy [155]. Several clinical trials of CAR-T therapy are currently ongoing [156].

## 7. ILD

Juvenile DM/DM-associated ILD is classified as CTD-ILD [157]. Although CTD-ILD is separated from idiopathic interstitial pneumonia (IIP), chest high-resolution computed tomography (HRCT) and the histopathological patterns of IIPs are commonly applied to CTD-ILD [157]. Both radiological and histopathological patterns are mostly compatible with non-specific interstitial pneumonia, organizing pneumonia, or a combination of both in adult DM; however, autopsy shows a diffuse alveolar damage pattern [158,159,160]. Although radiological and histopathological studies of ILD are limited in JDM, radiological patterns of non-specific interstitial pneumonia and organizing pneumonia have been reported in JDM-ILD [90]. IIP with clinical features that suggest an underlying autoimmune process, but do not meet established criteria for a CTD, is currently designated as “interstitial pneumonia with autoimmune features (IPAF)” [161]. In cases of respiratory symptoms due to ILD, screening for MSAs is helpful for the early diagnosis of an underlying JDM [161,162]. To date, many cases of fatal or severe ILD associated with JDM have been reported [90,116,117,118,119,144,145,146,147,148,149,152,162,163,164,165,166,167,168,169,170,171,172,173,174]. Interstitial lung disease is an early feature associated with mortality in JDM [66]. In contrast to adult cases, reports of ASS are rare [172,175]. Accordingly, most ILD cases, particularly in East Asia, are positive for anti-MDA5 autoantibodies [8,11,12]. The abrupt deterioration of respiratory failure suggests complications, such as pneumothorax or pneumomediastinum, which may develop secondary to ILD or vasculopathy [12,117,163,166].

### 7.1. Epidemiology

A French study demonstrated that 76% of patients with JDM showed a restrictive pattern of pulmonary function abnormalities, mostly attributed to respiratory muscle involvement [175]. A Norwegian study of 59 cases of JDM demonstrated low total lung capacity and diffusing capacity for carbon monoxide in 26% and 49% of cases, respectively; however, ILD was detected in 14% of cases using HRCT [176]. A study from Denmark demonstrated normal pulmonary function in 82% of patients with JDM and complications of ILD in 8% of patients [177]. Thus, screening for ILD using the pulmonary function test seems sensitive but not specific. The frequencies of ILD in JIIM are estimated to be 8% and 32.9% in a US and a Japanese cohort, respectively [3,9]. Given that pulmonary function testing is not available for young children and critically ill patients, chest HRCT should be performed, particularly in East Asia, where JDM-ILD is prevalent [14].

### 7.2. Laboratory Tests and Pathophysiology

Serum levels of Krebs von den Lungen-6 (KL-6) are useful markers for the diagnosis and monitoring of ILD, although they often remain moderately high despite apparent clinical and radiological improvements during CsA treatment [171]. High serum ferritin and IL-18 levels are associated with RP-ILD, suggesting that alveolar macrophages are involved in RP-ILD development in both adult and juvenile DM [12,178]. Involvement of alveolar macrophages is also suggested by elevated serum soluble CD163 levels in patients with anti-MDA5 antibody-positive adult DM-ILD [179,180]. Circulating neutrophil extracellular traps were significantly higher in anti-MDA5 antibody-positive adult patients with DM-ILD than those in antibody-negative patients, suggesting the involvement of neutrophils in ILD development [181]. Furthermore, Vβ genes of T cell receptors are selected and shared by the bronchoalveolar lavage fluid (BALF) and affected muscles in adult patients with DM-ILD, suggesting common target antigens in the lungs and muscles [182]. Flow cytometric analysis demonstrated increased numbers of CD8^+^ T cells in the BALF from DM/PM-ILD compared to scleroderma-ILD and healthy controls, suggesting a pathological role of cytotoxic T cells in DM/PM-ILD [183]. In contrast, the CD4^+^/CD8^+^ ratio in BALF was higher in patients with anti-MDA5-positive ILD than that in those with negative ILD [184]. A high CD4^+^/CD8^+^ ratio is observed in the peripheral blood of patients with acute/subacute ILD associated with anti-MDA5 antibody, despite the decreased number of circulating T cells [185]. Although analyses of serum cytokine profile demonstrated elevated IL-6, IL-8, TNF-α, and IP-10 in DM-ILD, anti-MDA5 antibody-positive cases showed higher serum IL-8 levels and lower IL-4/IFN-γ ratio than anti-MDA5 antibody-negative cases [186]. These findings suggest a distinct immunological profile for anti-MDA5 antibody-positive DM-ILD. Although MDA5 is expressed in the lungs of patients with both anti-MDA5 antibody-positive DM-associated ILD and IIPs, the deposition of immune complexes and complements is only detected in DM-ILD cases [187]. This suggests that MDA5 is nonspecifically overexpressed in the lungs of patients with ILD, and that anti-MDA5 autoantibodies exaggerate the progression of ILD. Indeed, the titers of anti-MDA5 autoantibodies are higher in the RP-ILD group than the chronic ILD group [11,12]. Additionally, serum levels of both B cell activating factor belonging to the tumor necrosis factor family (BAFF) and a proliferation-inducing ligand (APRIL) are significantly higher in the RP-ILD group than the chronic ILD group, suggesting that these cytokines are involved in the production of anti-MDA5 autoantibodies [188].

### 7.3. Treatment of JDM-ILD

We demonstrated the clinical efficacy of CsA in combination with IVMP for chronic and rapidly progressive ILD associated with JDM [90]. Another study reported the efficacy of IVCYC in four cases of JDM-associated ILD [173]. Because of the high frequency of anti-MDA5 antibody-positive ILD, early commencement of IVMP in combination with CNI and/or IVCYC is recommended in the Japanese practice guidance by the SRGRD [14]. However, there is no high-level evidence supported by a controlled study on the treatment of JDM-associated ILD, possibly because of the rarity of this disease. Thus, pediatricians should apply therapeutic strategies derived from studies on adult patients with DM to pediatric patients. A single-arm trial of GC + TAC combination therapy demonstrated a favorable 52 week-survival rate (82%) in adult patients with DM/PM-associated ILD [189]. Additionally, triple combination therapy with GC, TAC, and IVCYC dramatically improved the short- and long-term outcomes of DM-associated RP-ILD [65,190]. Although ILD complicates 80% of anti-MDA5 autoantibody-positive JDM, most cases do not progress rapidly and respond to high-dose GC combined with either CNI or IVCYC [9,12,90,173]. Our study of 10 cases of JDM-associated RP-ILD indicated that three survivors had no dry cough or dyspnea at the diagnosis of ILD, but received multi-drug therapy for ILD detected by HRCT [12]. Notably, three of seven fatal cases had no dyspnea and dry cough, and showed a mild subpleural curvilinear shadow or pleural effusion without ground-glass opacity at initial assessment. Nevertheless, they showed a rapidly progressive course and died of acute respiratory distress syndrome (pathologically confirmed as a diffuse alveolar damage pattern by autopsy) within four months after ILD diagnosis, despite aggressive treatment [12]. Thus, it is often difficult to predict the rapid progression of ILD at initial presentation [12]. The triple combination regimen should be initiated in patients with dyspnea, diffuse or multiple ground-glass shadows on initial CT, or high levels of anti-MDA5 autoantibodies. Additionally, the triple combination regimen should also be commenced in cases of ILD progression within one or two weeks despite high-dose GC combined with either CNI or IVCYC. A case report showed successful treatment with RTX in combination with extracorporeal membrane oxygenation and plasma exchange in a patient with JDM-associated severe RP-ILD that was refractory to triple combination therapy with IVIG [118]. A case series demonstrated the effectiveness of RTX in four of five patients with anti-MDA5 antibody-positive JDM-associated ILD; one patient died of respiratory failure that had progressed before the commencement of RTX [119]. Given the pathogenic role of anti-MDA5 antibodies [176], the elimination of autoantibodies and autoantibody-producing B cells may contribute to remission in refractory disease. A case series demonstrated the efficacy of tofacitinib in nine patients with JDM-associated refractory ILD, including three with RP-ILD [148]. Furthermore, JAK inhibitors have been used for the treatment of JDM-ILD refractory to conventional immunosuppressants, with or without biologics, and is a candidate for such cases [144,145,146,147,148,149,152,170].

## 8. Conclusions

Recent studies have provided new insights into the pathophysiology of JIIM. Myositis-specific autoantibodies are associated with distinct clinical phenotypes and outcomes. Accordingly, the inclusion of MSAs may improve the sensitivity and specificity of currently available classification criteria. Elucidation of the immunological aspects of JDM/JIIM supports the rationale for the use of new therapeutic agents, such as biologics and JAK inhibitors. Although RP-ILD remains a major cause of death, its outcome has improved with early diagnosis and intervention using a multi-drug regimen. Case reports and case series have demonstrated the successful treatment of patients who have already developed ARDS with JAK inhibitors in combination with or without conventional immunosuppressants. Therefore, RTX and JAK inhibitors are promising therapies for refractory JDM, including RP-ILD, in patients positive for anti-MDA5 antibodies.

## Figures and Tables

**Figure 1 children-11-01046-f001:**
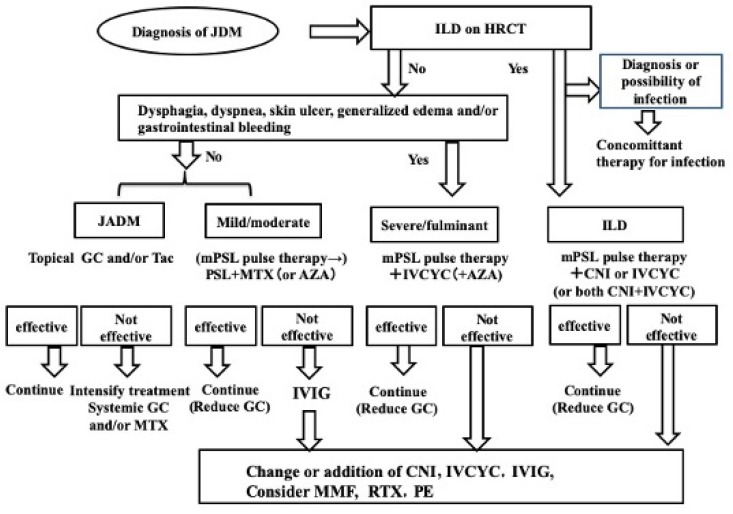
Algorithm of treatment for JDM (Ref. [14]). ADM, amyopathic dermatomyositis; AZA, azathioprine; CNI, calcineurin inhibitor; GC, glucocorticoid; HRCT, high-resolution computed tomography; ILD, interstitial lung disease; IVCYC, intravenous cyclophosphamide; IVIG, intravenous immunoglobulin; MMF, mycophenolate mofetil; mPSL, methylprednisolone; PE, plasma exchange; RTX, rituximab.

**Table 1 children-11-01046-t001:** Prevalence of myositis-specific autoantibodies.

	Total Number of Cases	ARS	Mi-2	TIF1-γ	MJ/NXP-2	MDA5	Negative	Ref.
Adult DM		9–24%	9–24%	13–31%	1–17%	Caucasian 0–13% Asian 10–48%		[54]
JDM (USA)	453	3.3%		34.7	25.6%	7.7%	13.2%	[7]
JIIM/UK	267	2%	4%	18%	15%	7%	41%	[6]
JDM (India)	34		2.9%			8.8%		[56]
JDM (Turkey)	56		8.7%	17.4%	21.7%	8.7%	24%	[57]
JDM (China)	76			11%	20%	17%	47%	[58]
JIIM/Hokkaido-Japan	25	8%	0%	16%	16%	28%	28%	[8]
JIIM/Japan	96	1%	4%	26%	26%	32%	11%	[9]

**Table 2 children-11-01046-t002:** Target antigens and associated clinical features of myositis-specific antibodies.

Myositis-Specific Antibodies	Target Antigen	Associated Clinical Features
Anti-CADM140 (MDA5) antibody	Melanoma-differentiation associated gene 5 (MDA5)	DM/JDM, RP-ILD, CADM in adult DM, arthritis, ulcer, inverse Gottron
Anti-p155/p140(TIF1-γ) antibody	Transcriptional intermediary factor 1-γ (TIF1-γ)	DM/JDM, typical and chronic/recurrent JDM, CAJDM, malignancy in adult DM
Anti-MJ/NXP-2 antibody	Nuclear matrix protein (NXP)-2	DM/JDM, severe myositis, high levels of serum CK, calcinosis, DM/JDM sine myositis
Anti-Mi-2 antibody	240/218 kDa helicase family protein	Classical DM/JDM
Anti-SAE antibody	Small ubiquitin-like modifier activating enzyme (SAE)	DM
Anti-SRP antibody	Signal recognition particle (SRP)	IMNM severe myopathy
Anti-3-hydroxy-3-methylglutaryl-coenzyme A reductase (HMGCR) antibody	3-hydroxy-3-methylglutaryl-coenzyme A reductase (HMGCR)	IMNM, severe myopathy, exposure to statins in adults
Anti-aminoacyl tRNA synthetase (ARS) antibody		Anti-symthetase syndrome; myositis, ILD, Raynaud phenomenon, mechanic’s hand

**Table 3 children-11-01046-t003:** Currently used “Guidance for diagnosis in Research Project of Specified Pediatric Chronic diseases” [14].

Guidance for Diagnosis
(1) Dermatological symptom
(a) Heliotrope rash: reddish purple edematous erythema in the eyelids, unilateral or bilateral (b) Gottron’s signs: Erythematous or violaceous macules on extensor surface of the finger joints with hyperkeratosis and dermatrophy (c) Erythema of extensor surfaces of the joints of the elbow, knee, etc. *^1^ (d) Findings of skin biopsy consistent with dermatomyositis *^2^
(2) Muscular symptom
Muscle weakness of proximal muscles in upper or lower extremities *^3^
(3) Imaging
Findings indicating myositis with MRI: High intensity on T2-weighted/fat suppression MRI and normal intensity on T1-weighted MRI
(4) Biochemical examination
Elevated serum level of muscle enzymes (creatine kinase or aldolase)
(5) Immunological examination
Positive result for myositis-specific autoantibodies *^4^
(6) Pathological examination
Pathological findings indicating myositis with muscle biopsy (degeneration of muscle fibers and cellular infiltration)
Diagnostic criteria (a) Classical JDM: The presence of one or more items of (1) dermatological symptoms (a) to (c), (2) muscular symptoms and two or more items of (3) to (6) during the follow-up. (b) JHDM: The presence of one or more items of classical dermatological symptoms (a) to (c), and one or more items of findings indicating myositis (3) to (6) without clinical muscle weakness. (c) JADM: The presence of one or more items of classical dermatological symptoms (a) to (c) without any evidences of myositis (2) to (6). (d) JPM: The presence of three or more items of (2) to (6) without dermatological symptoms (1).
Exclusions Myositis caused by infections, non-infectious myositis such as eosinophilic myositis, autoinflammatory diseases with rash similar to DM such as Nakajo–Nishimura syndrome, drug-induced myopathy, myopathy due to endocrine abnormality or congenital anomaly, muscle symptoms due to electrolyte abnormality, muscular dystrophy and other congenital muscular diseases, muscle weakness due to central or peripheral neuropathy, psoriasis, eczema, and other related diseases with other causes such as allergy.
Comments If a patient presents with following symptoms and/or complications, JDM should be considered. (1) Dermatological symptoms: erythema around the nail, erythema in anterior neck of upper chest (V-sign), erythema in shoulder to upper back (shawl sign), skin ulcer, Raynaud’s symptoms (2) Muscular symptoms: muscle pain, and muscle pain caused by pressure (3) Respiratory system: interstitial pneumonia (dry cough and exertional dyspnea in advanced cases), respiratory muscle weakness, and nasal voice (4) Gastrointestinal tract: dysphagia, and gastrointestinal ulcer and bleeding (5) Abnormalities on electrocardiograms (block, extra systole, changes in ST-T segment), myopathy, and pericarditis (6) Non-erosive arthritis (7) Frequent appearance of fever, malaise, weight loss and fatigability, which may rarely be accompanied by systemic edema (8) Dilatation and loop formation or loss of capillary vessels in the nail bed. These symptoms are not specific for JDM but are often found in JDM. (9) Calcinosis (skin, subcutaneous tissue, muscle/fascia, and bone/joint

Note: *^1^: Ulcerative lesions or secondary infection may modify the appearance of rash. *^2^: Hyperkeratosis, vacuolation of basal keratinocytes, deposition of melanin, perivascular lymphocyte infiltration, increased dermal edema, mucin deposition and thickening or atrophy of skin are observed, but it is difficult to differentiate JDM from SLE by pathological findings alone. Thus, a single symptom is not adopted as a dermatological finding. *^3^: Muscle weakness ranges from mild (e.g., stumbling, new onset of difficulties in exercise) to advanced (e.g., difficulties in standing up from sitting position or rolling over in bed). *^4^: Although only anti-Jo-1 antibodies could be measured at the time of revision of these guidelines, anti-ARS, anti-MDA5, anti-Mi-2, and anti-TIFI-γ antibodies are currently listed on the national health insurance price list. The other antibodies can be measured in some institutions but may be commercially available in the future.

## Data Availability

Not applicable.

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
