# Peer review of "Advances in Juvenile Dermatomyositis: Pathophysiology, Diagnosis, Treatment and Interstitial Lung Diseases—A Narrative Review"

_children, 2024, doi:10.3390/children11091046_

Round 1
Reviewer 1 Report
Comments and Suggestions for Authors
The Manuscript: 'Juvenile dermatomyositis and associated interstitial lung diseases: advances in pathophysiology, diagnosis, and treatment', by Ichiro Kobayashi, delves into the pathophysiology, clinical manifestations, and therapeutics of juvenile idiopathic inflammatory myopathy. After carefully reviewing the manuscript, I consider that the manuscript is of high scientific quality.
Author Response
Thanks so much for your favorable comments.
Reviewer 2 Report
Comments and Suggestions for Authors
Dear Author!
Thank you for the opportunity to review your manuscript.
IIM in children are rare group of the diseases with very complex pathogenesis including autoimmunity and autoiflammatory (interferonopathy) mechanisms
The manuscript is a detailed review about all aspects of this disease, including classification, clinics, diagnostic criteria and role of each antibody, genes and interferonopathic mechanisms in the pathogenesis of the disease.
Manuscript contains all known information about the treatment options
Despite the title about ILD, associated with IIM this part is a relatively small and manuscript is more about all aspects of IIM than ILD.
I think the more reliable title is about all aspects of IIM, not only ILD
The manuscript contains all known literature about IIM in children
Author Response
Thanks so much for your comments.
According to your advice, I have changed the title to “Advances in juvenile dermatomyositis: pathophysiology, diagnosis, treatment and interstitial lung diseases– a narrative review–”.
Reviewer 3 Report
Comments and Suggestions for Authors
In this review, the Author reviewed the main issue about JIIM. The title is quite informative while the main text is exhaustive. Tables are very informative. In general the paper is very interesting even if the disease is very rare. However a paper like this can be very helpful for clinicians.
Minor comments:
- Title: please specify if this is a systematic or narrative review (in the first case report it in a Methods section)
- Table 1. What does they mean the numbers in the second column?
- Consider to better organize with subheadings the sections 6 and 7.
Author Response
Thanks so much for your comments.
Comment: Title: please specify if this is as systematic or narrative review
Reply: Thank you for your comment. This is a narrative review. I added this in the title “Advances in juvenile dermatomyositis: pathophysiology, diagnosis, treatment and interstitial lung diseases– a narrative review–”.
Comment; Table 1: What does they mean the numbers in the second column?
Reply: The column means the number of cases included in each study. I added this in the table.
Comment: Consider to better organize with subheadings the section 6 and 7.
Reply: Thank you for your advice. I added subheadings in section 6 and 7 as below.
Section 6: Management
6.1. Principal of treatment
6.2. Conventional immunosuppressants
6.3. Intravenous immunoglobulin
6.4 Biologics
6.5. JAK inhibitors
6.6. Others
Section 7: ILD
7.1. Epidemiology
Also, I have added the following sentence in this section “ The frequencies of ILD in JIIM are estimated to be 8% and 32.9% in a US and a Japanese cohort, respectively [3,9]”.
7.2. Laboratory tests and pathophysiology
7.3. Treatment of JDM-ILD
Round 2
Reviewer 3 Report
Comments and Suggestions for Authors
All comments were addressed